# A deeper analysis in thyroid research: A meta-epidemiological study of the American Thyroid Association clinical guidelines

**Dalia A. Castillo-Gonzalez**[1,2☺], **Edgar G. Dorsey-Trevino**[1,2☺], **Jose G. Gonzalez-Gonzalez**[1,2,3], **Mariana Garcia-Leal**[1], **Karen G. Bautista-Orduño**[1,2], **Karina Raygoza**[2], **Michael R. Gionfriddo**[4], **Naykky M. Singh Ospina**[5,6], **Rene Rodriguez-Gutierrez**[ID][1,2,5,7] *

1 Endocrinology Division, Department of Internal Medicine, University Hospital "Dr. José E. González", Universidad Autónoma de Nuevo León, Monterrey, México, 2 Plataforma INVEST Medicina UANL–KER Unit Mayo Clinic (KER Unit México), Universidad Autónoma de Nuevo León, Monterrey, México, 3 Research Unit, University Hospital "Dr. José E. González", Universidad Autónoma de Nuevo León, Monterrey, México, 4 Center for Pharmacy Innovation and Outcomes, Geisinger Health System, Forty Fort, PA, United States of America, 5 Knowledge and Evaluation Research Unit in Endocrinology, Mayo Clinic, Rochester, MN, United States of America, 6 Division of Endocrinology, Department of Medicine, University of Florida, Gainesville, FL, United States of America, 7 Division of Endocrinology, Diabetes, Metabolism, and Nutrition, Department of Medicine, Mayo Clinic, Rochester, MN, United States of America

☺ These authors contributed equally to this work.
* rodriguezgutierrez.rene@mayo.edu

## Abstract

### Background

The American Thyroid Association (ATA) uses the GRADE or the American College of Physicians (ACP) system to develop recommendations. Recommendations based on low-quality evidence should spur for the conduction of clinical studies, if feasible. The extent to which recommendations by the ATA based on low-quality of evidence are being actively researched remains unknown.

### Methods

Clinical guidelines produced by the ATA using the GRADE or the ACP system to classify evidence were deemed eligible. Reviewers, in duplicate and independently, extracted therapeutic recommendations based on low-quality evidence, whereas recommendations with higher quality of evidence, aimed at diagnosis, or best practice statements were excluded. Eligible recommendations based on low-quality evidence were deconstructed to their components using the PICO format. We then searched on clinicaltrials.gov to identify ongoing research. Trials were deemed eligible if they addressed the PICO question with at least one of the intended outcomes.

### Results

A total of 543 recommendations were retrieved, of which 305 (56%) were based on low-quality of evidence and only 90 were deemed eligible. Of these, we found that 33 (37%) recommendations were actively being researched in 53 clinical trials. Most of the trials were

**Data Availability Statement:** All relevant data are within the paper and its Supporting Information files.

**Funding:** The authors received no specific funding for this work.

**Competing interests:** The authors have declared that no competing interests exist.

randomized and funded by non-profit organizations. Many clinical trials studied thyroid nodules and differentiated thyroid cancer (26/53; 49%), whereas few studied were aimed at anaplastic thyroid cancer (2/53; 4%).

## Conclusion

One out of three of gaps in evidence, identified as low quality during the development of ATA guidelines, are currently actively researched. This finding calls for the need to develop a better research infrastructure and funding to support thyroid research.

## Introduction

When creating guideline recommendations, guideline panelists estimate the benefits and harms of relevant options and rate the confidence they have in those estimates. [1, 2] The Grading of Recommendations Assessment, Development and Evaluation (GRADE) approach facilitates this process by offering guidance about how to determine the quality of evidence and strength of each recommendation. [3, 4] Quality of the evidence is determined to be low or very-low when the source of the estimates is susceptible to bias or methodological limitations (e.g., observational studies), [5] when several studies provide widely different estimates of effect (i.e., inconsistency), [6] or the estimates are not directly applicable to the population of interest (i.e., indirectness) [7]. As a result, recommendations derived from low or very-low quality of evidence lead to weak (or conditional) recommendations that are often seen as gaps in knowledge. [8, 9]

These knowledge gaps present areas of opportunities for research that will likely have an impact on the quality of evidence, strength of future recommendations, and patient care. However, recent evidence suggests that these knowledge gaps often remain so due to lack of active research. [10] Recently, the American Thyroid Association (ATA)—a professional organization that develops clinical practice guidelines for the care of patients with thyroid conditions—has crafted some clinical guidelines based on the GRADE system and the American College of Physicians (ACP) Guideline Grading System, which is a modified version of the GRADE system that adopted most of its rationale and judgmental criteria when developing clinical recommendations [11]. Since then, five clinical guidelines have been developed based on these systems—hyperthyroidism and thyrotoxicosis, [12] differentiated thyroid cancer and thyroid nodules, [13] thyroid dysfunction in pregnancy and postpartum, [14] hypothyroidism, [15] and anaplastic thyroid cancer. [16] There is, therefore, an opportunity to assess current knowledge gaps and the current research environment within conditions affecting the thyroid.

To that end, we performed a meta-epidemiological study on ATA guidelines aiming to describe recommendations supported by low-quality evidence and determine the extent to which these potential knowledge gaps are being actively researched.

## Methods

### Study design

This study was conducted in alignment with an existing reporting guideline for meta-epidemiological studies (S1 Table). [17] A meta-epidemiological study is a synthesis of non-clinical evidence that usually uses non-patient outcomes, such as specific characteristics or settings of clinical studies, describing distribution of evidence, examining heterogeneity and exploring its

causes, identifying and describing plausible biases, and providing empirical evidence for hypothesized association. [17]

### Eligibility criteria and data collection

Clinical guidelines were downloaded from the ATA web page (https://www.thyroid.org/professionals/ata-professional-guidelines/) and only guidelines using the GRADE or the ACP system to provide recommendations were selected. [12–14, 16, 18] Using a standardized web-extraction form (Microsoft Excel 2016, Microsoft®, Redmont, WA, USA), reviewers, in duplicate and independently, extracted; guideline topic, year of publication, strength of recommendation, and quality of evidence. A pilot test using 20 recommendations was performed to standardized reviewers' criteria and avoid misclassification of recommendations as either diagnostic, prognostic, or intervention. Recommendations about treatment interventions based on low-quality of evidence were included, whereas those based on moderate or high-quality evidence were excluded. Recommendations focused on either diagnostic or prognostic evaluations, although commonly associated with low-quality evidence, were purposefully excluded as these studies are less likely to be answered with or conducted with an interventional study. [19] Also, there are some recommendations for which no sensible alternative exists, and thus, no one would consider doing a study to elucidate the answer to the implicit question. These recommendations were labeled as "best practice" and were also excluded from the study (e.g., *patients should be informed of side effects of ATDs and the necessity of informing the physician promptly if they should develop pruritic rash, jaundice, acholic stools or dark urine, arthralgias, abdominal pain, nausea, fatigue, fever, or pharyngitis. Preferably, this information should be in writing. Before starting ATD and at each subsequent visit, patient should be alerted to stop the medication immediately and call their physician if there are symptoms suggestive of agranulocytosis of hepatic injury*). [20] We have provided a table with examples of the aforementioned recommendations (S2 Table). Clinical guidelines using the ACP system lack de classification of "best practice, but as this system mimics the rationale and criteria of the GRADE system to craft recommendations, experts in the GRADE system (RR-G and MRG) analyzed all recommendations and determined which recommendations were "best-practice".

To elucidate the research gaps, recommendations having low-quality of evidence were deconstructed into its components using the PICO format—patient, intervention, comparison, and outcome—in order to create a research question for which a clinical trial could plausibly answer and to ensure the reproducibility of our methods. To calibrate reviewers' criteria for the latter, a pilot test using 15 recommendations were used to depict the research question, yielding an initial agreement of 80%. This process was repeated until percentage agreement was of 100%; this was achieved after 25 recommendations. Disagreements were initially resolved by consensus, and whenever this was not possible, an expert endocrinologist and/or methodologist (RR-G or MRG) made the decision.

A search on clinicaltrials.gov—considered the most complete registry site for clinical trials as it includes 208 countries—was performed. We used the research questions obtained from the recommendations based on low-quality of evidence for our search. [21] We specifically looked for clinical trials and excluded observational studies as they often provide low-quality evidence. A clinical trial was deemed eligible if it addressed the PICO question with at least one of the intended outcomes and if they were either completed or ongoing less than 5 years prior to the publication of the respective clinical guideline. Extracted information from each study included; funding, year of registry, location of study (country and continent), number of centers, type of allocation, recruitment status (i.e. not yet enrolling, ongoing, active),

intervention model (e.g., single group, parallel, crossover, or factorial assignment), phase (i.e., I-IV), and masking (blinding).

To record the adherence from each clinical trial with the formulated research question, we labeled each trial into complete, partial, or incomplete alignment. Trials with complete alignment had to match all four dimensions of the PICO question, whereas trials with ≤2 dimensions were deemed to have incomplete alignment. All other trials were labeled as partially aligned. Reviewers working independently searched the clinical trial registry until a 100% agreement was achieved, a point reached after searching for 20 questions. A summarized diagram of the research methods is depicted in Fig 1.

## Statistical analysis

We used descriptive statistics to report categorical variables with frequencies and percentages. We used SPSS version 22 (IBM Corp®, Armonk, NY, USA) for all statistical analysis. Graphics were designed using Microsoft Excel 2016 (Microsoft®, Redmont, WA, USA).

## Results

We retrieved a total of seven clinical guidelines from the ATA published between 2012 to 2017, of which only five used the GRADE system to craft recommendations (Fig 2). [12–16] Interobserver agreement in the extraction of recommendations ranged from 0.9 to 1 and yielded a total of 543 recommendations. High-quality evidence supported 7% (n = 39) of the recommendations, moderate-quality 37% (n = 199), low-quality 51% (n = 278), and 5% (n = 27) of the recommendations were labeled as having insufficient evidence to make recommendations (Fig 2). A total of 238 (44%) recommendations were excluded due to moderate- or high-quality evidence. Afterward, recommendations considered as best practice or aimed at diagnostic endpoints were excluded, leaving a total of 90 recommendations supported by low-quality evidence (Fig 2).

Less than half of these recommendations (33/90; 37%) are being actively researched (Fig 3). Current active research is primarily focused on differentiated thyroid cancer (DTC) (14 out of 24 recommendations; 58%) and hypothyroidism (7 out of 14 recommendations; 50%); anaplastic thyroid cancer was the least active area of active research. (ATC) (1 out of 5; 16%) (Fig 3).

Most of the trials were funded by non-profit organizations (47/53; 89%), had a parallel design (30/53; 57%), were randomized (32/53; 60.4%), and were actively recruiting (21/53; 40%) (Table 1). Almost two-thirds of trials (33/53; 62%) were found to be in complete alignment with the PICO question, whereas only 3 of 53 trials (6%) had an incomplete alignment (Table 1).

## Discussion

### Summary of findings

Less than 10% of the recommendations endorsed by the ATA guidelines stem from high-quality evidence while over 50% is based on low-quality evidence. Only one out of three knowledge gaps identified in management recommendations of common thyroid disorders are actively being researched.

### Comparison with previous studies

Our findings align with previous evidence elucidating the extent of which clinical guidelines are based on low-quality of evidence. [22, 23] Murad et. al. determined that 65% of

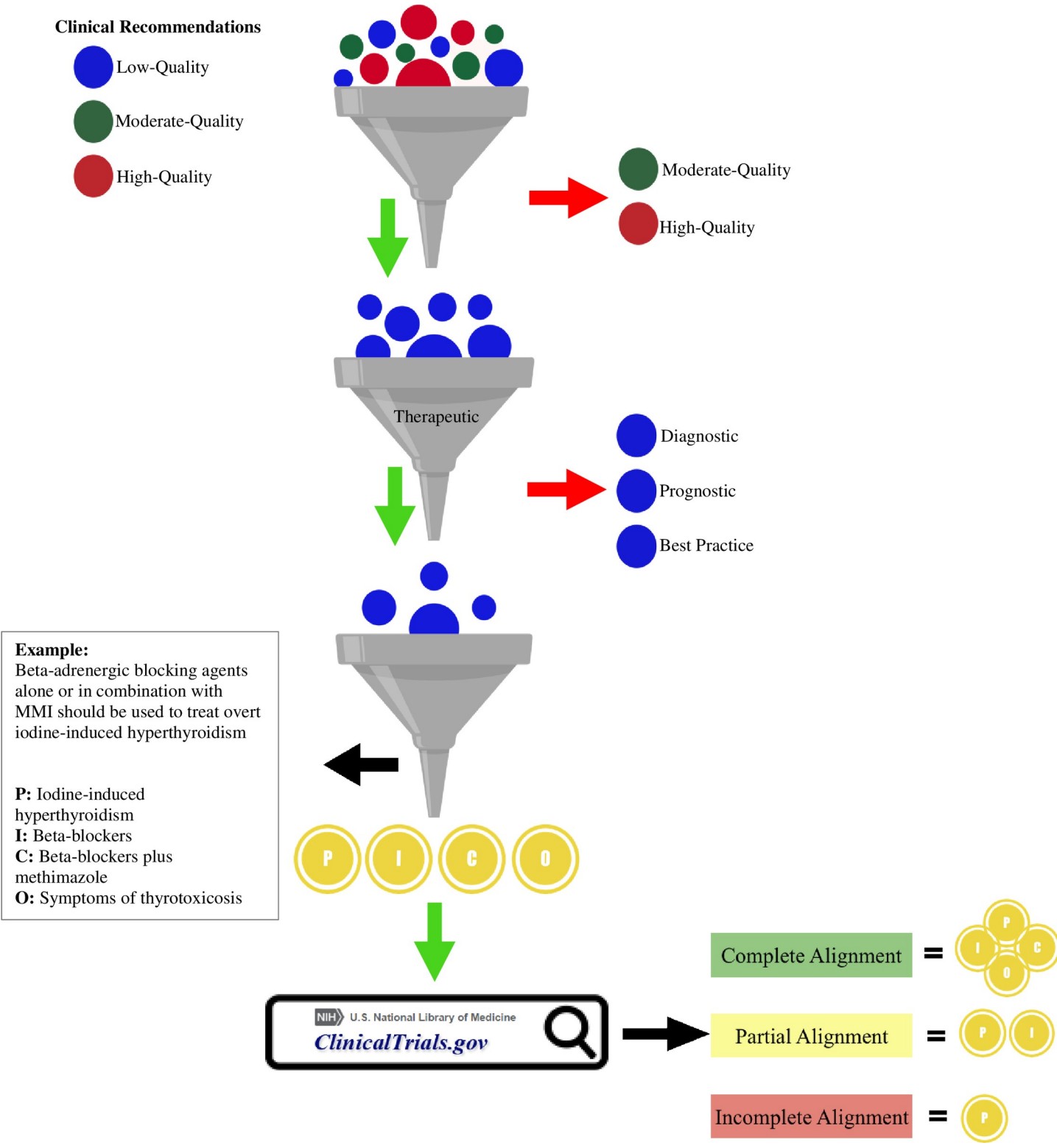

**Fig 1. Summary of methods.** Balls represent clinical recommendations. Red arrows mean exclusion criteria, whereas green arrows means inclusion criteria. P: Patient; I: Intervention; C: Comparison; O: Outcome.

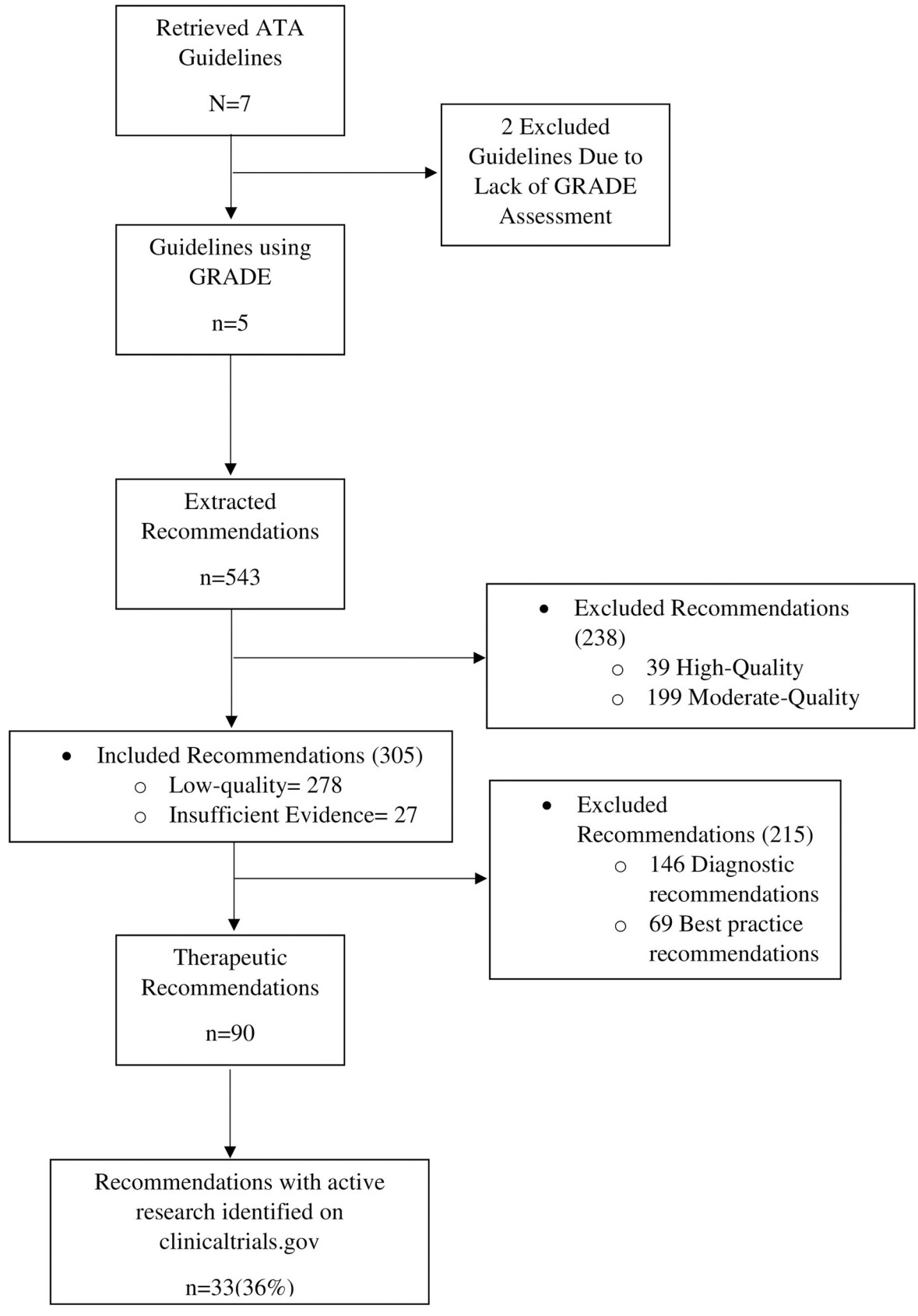

**Fig 2. Flow diagram of included guidelines and recommendations.**

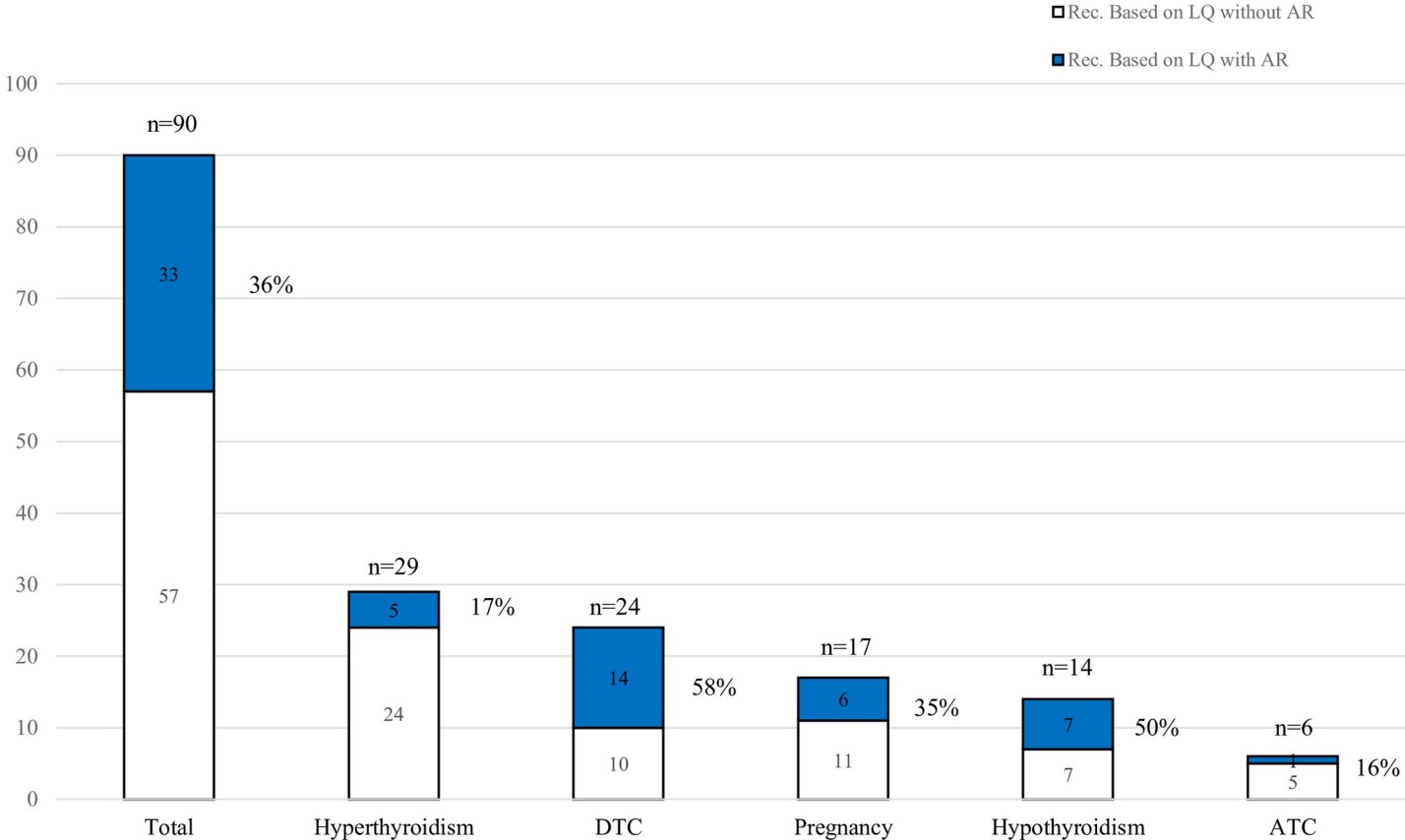

**Fig 3. Proportion of recommendations based on low-quality evidence with and without active research.** Percentages represents recommendations with low-quality of evidence with active research. LQ: Low-quality; Rec: Recommendation; AR: Active Research; DTC: Differentiates Thyroid Cancer; ATC: Anaplastic Thyroid Cancer.

recommendations crafted by the Society for Vascular Surgery were strong but based on low-quality of evidence. [24] Similar findings have been found in the American College of Cardiology/American Heart Association and the Infectious Diseases Society of America guidelines in which nearly 48% and 50% of recommendations were drawn based on low-quality of evidence, respectively. [25, 26] More recently, an assessment made by Alexander et. al. on the World Health Organization (WHO) guidelines find that more than half (55.5%) of strong recommendations were based on either very- or low quality of evidence. [27] In endocrine, Brito et. al. evaluated the Endocrine Society guidelines and found that 58% of recommendations were predominantly based on observational studies and thus, conferring low-quality of evidence. [28] Conversely, an analytical survey performed by Agoristas et. al. showed that most recommendations spawned in UpToDate were based on moderate- or high-quality evidence. [29] None of the aforementioned studies, however, sought to determine the extent to which these knowledge gaps were being assessed by the research enterprise.

An analytical assessment performed by Checketts et. al. on the American Society of Plastic Surgeon's clinical guidelines for breast reconstruction found that 6 out of 10 identified research gaps were being evaluated by active research. [30] While, Singh Ospina et. al. found that 3 out of 10 recommendations based on low-quality of evidence the Endocrine Society guidelines were being assessed by active clinical trials. [10] Our findings are more in line with those of Singh-Ospina; we found that for the ATA guidelines, active research was being conducted on around one-third of the recommendations based on low-quality evidence.

**Table 1. Description of interventional studies.**

| Trial Characteristics | | Included Trials (53) |
|---|---|---|
| Thyroid Topic | | |
| | Pregnancy | 9 (17) |
| | Hyperthyroidism | 5 (9) |
| | TN and DTC | 26 (49) |
| | Hypothyroidism | 11 (21) |
| | ATC | 2 (4) |
| Funding | | |
| | Non-Profit Sources | 47 (89) |
| | Industry | 6 (11) |
| Continent | | |
| | America | 20 (38) |
| | Europe | 20 (38) |
| | Africa | 3 (6) |
| | Asia | 10 (19) |
| Intervention Masking | | |
| | Parallel | 30 (57) |
| | Single Group | 16 (30) |
| | Factorial | 3 (6) |
| | Crossover | 4 (7) |
| Allocation | | |
| | Not Randomized | 7 (13) |
| | Randomized | 32 (60) |
| | No reported | 13 (24) |
| Phase | | |
| | 1 | 4 (7) |
| | 2 | 12 (23) |
| | 2 & 3 | 5 (9) |
| | 3 | 16 (30) |
| | 4 | 7 (13) |
| | N/A | 9 (17) |
| Status | | |
| | Active, Not Recruiting | 12 (23) |
| | Recruiting | 21 (40) |
| | Completed | 20 (38) |
| Alignment | | |
| | Complete | 33 (62) |
| | Partial | 17 (32) |
| | Incomplete | 3 (6) |

Data is presented as frequencies (percentages). TN: Thyroid Nodule; DTC: Differentiated Thyroid Cancer; ATC: Anaplastic Thyroid Cancer; N/A: Not Applicable

## Strengths and limitations

We utilized a protocol-driven method, that facilitates reproducibility however, our results have several limitations that may reduce the confidence in our estimates. First, there are several instances in which performing a clinical study may be unethical or not feasible and by excluding observational evidence, our results may underestimate the extent to which research is

addressing current knowledge gaps. However, we purposefully excluded best-practice recommendations which reduce this possibility. Second, we excluded other trial registries but due to the scope of clinicaltrials.gov we are confident that excluding additional registries did not appreciably alter our estimates. Additionally, most guidelines were using the ACP system to qualify recommendations, and thus, lacked a "best practice" category. But as this system mimics the rationale and criteria of the GRADE system and experienced methodologists made the final evaluation, we consider this limitation to be insignificant. Despite these limitations, this approach has successfully been used previously and experts in methodology and clinical research planned and guided the conduct of this study, [10] thereby supporting the validity of this approach.

## Implications for clinical practice and research

Based on our findings, we have found that most clinical recommendations, put forth by the ATA guidelines, are based on observational studies. This high prevalence may be due to several perks of thyroid research. For instance, conducting a multinational RCT requires for a multidisciplinary team that helps with the design, recruitment, analysis, and publication, and all these can only be afforded by proper funding. Yet, recent evidence has shown that thyroid research lacks the proper funding when compared to other areas with more prevalence and impact on morbidity and mortality. [31] This lack of funding impairs the proper conduction of RCT's, and therefore, limits panelists to utilize the best available evidence to create clinical guidelines, despite the best stands for mostly being observational studies. But are observational studies always deemed to deploy low-quality of evidence? The answer is no.

There is an unquestionable valuableness of observational studies to yield causal results whenever conducted properly and adjusting for possible confounders. [32, 33] Using observational data to make causal statements, however, requires the proper methods; causal questions requires causal method to answer them. For this, a rogue group of statisticians and epidemiologists defied the adage of "association is not causation" and stated that causality could be estimated from observational studies when proper methods are applied. [34–37] When this premises are violated, however, observational studies may yield an association between a treatment and an outcome even if this physiological plausibility fails to hold in real-world scenarios. For instance, one of the most famous cases that bespeak the latter concept was the use of hormone replacement therapy (HRT) to prevent CVD in postmenopausal women. [38] The Framingham Cohort Study showed that the incidence of myocardial infarction (MI) among premenopausal women was virtually inexistent. This led to the assumption that by treating postmenopausal women with HRT, the incidence of MI will decrease. [38] Yet, this surmise failed when tested in a RCT. [39] Along this example, there are several other studies that highlight the precautions of using observational studies to provide care when analyzed incorrectly. [39, 40]

On the other hand, the conduction of an RCT's are far from being exempt of bias. [41] Blindly inputting our faith towards everything that is labeled "randomized clinical trial" spurs for an inadequate application of evidence-based medicine to patient care. [42] A thorough and peruse appraisal of the evidence is necessary for clinicians to make confident clinical decisions and for panelists to develop trustworthy clinical recommendations. Additionally, for some thyroid condition, especially rare diseases, with low prevalence, or high mortality, conducting a RCT would demand a tremendous effort, funding, recruitment, and some sort of fortunateness, and most likely, the resulting RCT will be underpowered, unrepresentative, and with misleading results. Observational studies may sometimes be the best and most feasible option to provide a causal inference and implement it to patient care.

Our results have also denoted that even for diseases with high prevalence, such as hyperthyroidism, hypothyroidism, thyroid disease during pregnancy, and differentiated thyroid cancer, this knowledge gap is undesirably prevalent. In best case scenario, only half of these recommendations were being evaluated by a RCT's. A conjecture of our own is that the research enterprise has probably deemed these conditions as being of low priority due to their benign and relatively good prognosis. For instance, the 10-year mortality for papillary thyroid cancer has been estimated to be up to 97% in some studies, [43] whereas for hypo- and hyperthyroidism the prognosis are also considered to be fair when treated. Nonetheless, good prognosis is unrelated to a decrease in patients' expenditures and healthcare costs or an avoidance of the hindrance of life-long treatments or the permanent scarring of surgery. *De facto*, the increasing prevalence of these conditions would be putative of causing billion dollars in healthcare costs in the following years. [31]

## Opportunities for future research and enhancement of clinical guidelines

Beyond identifying knowledge gaps, we identified redundancy in some areas of research. While multiple studies on similar topics can be useful to explore heterogeneity and decrease imprecision, it may also be an indicator of misusage of resources. In efforts to avoid this, researchers may guide their research pipeline by using systematic reviews to justify new studies. [44–46] Furthermore, some of the ATA guidelines have highlighted areas of opportunities in which future research should direct their efforts. This section was found in 3 out of 5 guidelines (thyroid nodules and differentiated thyroid cancer, thyroid disease during pregnancy and postpartum, and hypothyroidism), [13–15] and we encourage panelists to add this same section for all other guidelines as it poses a trustable guidance for researchers to direct their efforts towards meaningful clinical queries.

Lastly, the ATA, as mentioned previously, primarily uses the ACP system to develop recommendations. Although a recent statement mentioned that there are piloting with the GRADE system, [47] full endorsement would aid in the advocacy of transparency and informativeness. For instance, the rating of the evidence in GRADE has 4 categories (very low, low, moderate, and high), whereas the ATA guidelines only used 3 (low, moderate and high). [4, 11] While this may simplify rating for clinicians, it may also obscure the most necessary areas of research.

## Conclusions

Less than one-third of low quality recommendations are actively being researched. Our results suggest that these guidelines may serve as an opportunity for the research enterprise to aim their research, after evaluating the feasibility, funding, and priorities, at these knowledge gaps.

## Supporting information

**S1 Table. Reporting guidelines for meta-epidemiological studies.**
(DOC)

**S2 Table. Paradigmatic situations in which panels may reasonably offer strong based on low confidence in effect estimates.**
(DOCX)

## Author Contributions

**Conceptualization:** Naykky M. Singh Ospina, Rene Rodriguez-Gutierrez.

**Data curation:** Dalia A. Castillo-Gonzalez, Edgar G. Dorsey-Trevino, Mariana Garcia-Leal, Karen G. Bautista-Orduño, Karina Raygoza, Rene Rodriguez-Gutierrez.

**Formal analysis:** Dalia A. Castillo-Gonzalez, Edgar G. Dorsey-Trevino.

**Funding acquisition:** Jose G. Gonzalez-Gonzalez.

**Investigation:** Jose G. Gonzalez-Gonzalez, Mariana Garcia-Leal, Karen G. Bautista-Orduño, Karina Raygoza, Rene Rodriguez-Gutierrez.

**Methodology:** Dalia A. Castillo-Gonzalez, Edgar G. Dorsey-Trevino, Jose G. Gonzalez-Gonzalez, Mariana Garcia-Leal, Karen G. Bautista-Orduño, Karina Raygoza, Rene Rodriguez-Gutierrez.

**Supervision:** Rene Rodriguez-Gutierrez.

**Validation:** Rene Rodriguez-Gutierrez.

**Visualization:** Rene Rodriguez-Gutierrez.

**Writing – original draft:** Dalia A. Castillo-Gonzalez, Edgar G. Dorsey-Trevino.

**Writing – review & editing:** Edgar G. Dorsey-Trevino, Jose G. Gonzalez-Gonzalez, Michael R. Gionfriddo, Naykky M. Singh Ospina, Rene Rodriguez-Gutierrez.

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
