## [Decision Letter · Decision Letter 0]

8 May 2020

PONE-D-19-35572

A Deeper Analysis in Thyroid Research: A Meta-Epidemiological Study of the American Thyroid Association Clinical Guidelines

PLOS ONE

Dear Dr. Rodriguez-Gutierrez

Thank you for submitting your manuscript to PLOS ONE. Our sincere apologies for the longer than usual time taken to review this manuscript. We had a challenge with getting the most appropriate feedback. After careful consideration, we feel that it has merit but does not fully meet PLOS ONE’s publication criteria as it currently stands. Therefore, we invite you to submit a revised version of the manuscript that addresses the points raised during the review process.

We would appreciate receiving your revised manuscript by Jun 22 2020 11:59PM. To enhance the reproducibility of your results, we recommend that if applicable you deposit your laboratory protocols in protocols.io, where a protocol can be assigned its own identifier (DOI) such that it can be cited independently in the future. For instructions see: http://journals.plos.org/plosone/s/submission-guidelines#loc-laboratory-protocols

We look forward to receiving your revised manuscript.

Kind regards,

Honey V. Reddi

Academic Editor

PLOS ONE

Journal Requirements:

Reviewers' comments:

Reviewer's Responses to Questions

**Comments to the Author**

1. Is the manuscript technically sound, and do the data support the conclusions?

Reviewer #1: Yes

Reviewer #2: Partly

2. Has the statistical analysis been performed appropriately and rigorously? 

Reviewer #1: Yes

Reviewer #2: Yes

3. Have the authors made all data underlying the findings in their manuscript fully available?

Reviewer #1: Yes

Reviewer #2: Yes

4. Is the manuscript presented in an intelligible fashion and written in standard English?

Reviewer #1: Yes

Reviewer #2: Yes

5. Review Comments to the Author

Reviewer #1: This is an interesting and well written manuscript that highlights the reality of difficulties in reliance in evidence-based medicine and how much of what clinical guidelines (in almost every field of Medicine) recommend is based on less than optimal evidence and the need to constantly review and revise guidelines as new evidence emerges.

It would be very helpful (even if not the original intent of this paper) for the authors to highlight 3 specific areas or questions/recommendations in the field of thyroidology which would benefit from realistic RCT that would help advance the field. Many questions are not easily answered, others cannot be subjected to RCT for ethical reasons, etc.

Reviewer #2: The authors have examined the American Thyroid Association guidelines (those which use the ACP system to classify evidence) to identify those recommendations with low-quality evidence and determine which of these are being researched. Of the 5 sets of guidelines, 56% were based on low-quality evidence and 90 were further examined. Overall, 37% were actively being researched in clinical trials, suggesting gaps in thyroid research and a call for consideration of increased funding.

Comments:

1. It would be helpful to have a supplemental table/index of those recommendations that the authors found to be 'best practices' or for 'diagnostic' purposes. I was not clear from their methodology what they meant by that term and how it reflected in the guidelines chosen.

2. The authors discuss previous studies that have examined the pairing of strength of recommendation with quality of evidence; perhaps the authors could also elucidate which of the low-quality guidelines were associated with what strength of recommendation and if this correlated to the presence (or absence) of clinical trials;

3. the authors encourages panelists on guidelines to fully endorse the GRADE guidance. I do not think this is appropriate in the context of this manuscript.

6. PLOS authors have the option to publish the peer review history of their article (what does this mean?). If published, this will include your full peer review and any attached files.

Reviewer #1: No

Reviewer #2: No

---

## [Author Response · Author response to Decision Letter 0]

12 May 2020

Reviewer #1

Comment #1:

This is an interesting and well written manuscript that highlights the reality of difficulties in reliance in evidence-based medicine and how much of what clinical guidelines (in almost every field of Medicine) recommend is based on less than optimal evidence and the need to constantly review and revise guidelines as new evidence emerges.

It would be very helpful (even if not the original intent of this paper) for the authors to highlight 3 specific areas or questions/recommendations in the field of thyroidology which would benefit from realistic RCT that would help advance the field. Many questions are not easily answered, others cannot be subjected to RCT for ethical reasons, etc

Response:

We appreciate the kind words and insightful observation of the reviewer. We recognize that the conduction of RCTs may be unfeasible due to funding, ethics, or other reasons, and that most of the times, the use of observational data is what is available as best evidence. Although it is impossible for this paper to provide a comprehensive review of current epidemiologic methods to draw causal statements from observational data, we have added a paragraph and several references that will guide the readers in case they are interested.

The manuscript reads as follows:

Based on our findings, we have found that most clinical recommendations, put forth by the ATA guidelines, are based on observational studies. This high prevalence may be due to several perks of thyroid research. For instance, conducting a multinational RCT requires for a multidisciplinary team that helps with the design, recruitment, analysis, and publication, and all these can only be afforded by proper funding. Yet, recent evidence has shown that thyroid research lacks the proper funding when compared to other areas with more prevalence and impact on morbidity and mortality.[31] This lack of funding impairs the proper conduction of RCT’s, and therefore, limits panelists to utilize the best available evidence to create clinical guidelines, despite the best stands for mostly being observational studies. But are observational studies always deemed to deploy low-quality of evidence? The answer is no. 

There is an unquestionable valuableness of observational studies to yield causal results whenever conducted properly and adjusting for possible confounders.[32, 33] Using observational data to make causal statements, however, requires the proper methods; causal questions requires causal method to answer them. For this, a rogue group of statisticians and epidemiologists defied the adage of “association is not causation” and stated that causality could be estimated from observational studies when proper methods are applied.[34-37] When this premises are violated, observational studies may yield an association between a treatment and an outcome even if this physiological plausibility fails to hold in real-world scenarios. For instance, one of the most famous cases that bespeak the latter concept was the use of hormone replacement therapy (HRT) to prevent CVD in postmenopausal women.[38] The Framingham Cohort Study showed that the incidence of myocardial infarction (MI) among premenopausal women was virtually inexistent. This led to the assumption that by treating postmenopausal women with HRT, the incidence of MI will decrease.[38] Yet, this surmise failed when tested in a RCT.[39] Along this example, there are several other studies that highlight the precautions of using observational studies to provide care when analyzed incorrectly.[39, 40] 

On the other hand, the conduction of an RCT’s are far from being exempt of bias.[41] Blindly inputting our faith towards everything that is labeled “randomized clinical trial” spurs for an inadequate application of evidence-based medicine to patient care.[42] A thorough and peruse appraisal of the evidence is necessary for clinicians to make confident clinical decisions and for panelists to develop trustworthy clinical recommendations. Additionally, some thyroid condition, especially rare diseases, with low prevalence, or high mortality, conducting a RCT would demand a tremendous effort, funding, recruitment, and some sort of fortunateness, and most likely, the resulting RCT will be underpowered, unrepresentative, and with misleading results. Observational studies may sometimes be the best and most feasible option to provide a causal inference and implement it to patient care.

Reviewer #2

Comment #1 

It would be helpful to have a supplemental table/index of those recommendations that the authors found to be 'best practices' or for 'diagnostic' purposes. I was not clear from their methodology what they meant by that term and how it reflected in the guidelines chosen.

Response:

We have added a table as a supplementary file from a previous publication in which we provide examples from each case. 

Comment #2

The authors discuss previous studies that have examined the pairing of strength of recommendation with quality of evidence; perhaps the authors could also elucidate which of the low-quality guidelines were associated with what strength of recommendation and if this correlated to the presence (or absence) of clinical trials.

Response

We appreciate the reviewer comment. However, classification based on the underlying quality was made solely on the recommendations, not the guidelines. Most guidelines had a mixed between low-, moderate-, and high-quality of evidence. We, however, acknowledge that it is important to elucidate what proportion of low-, moderate-, and high-quality recommendations had each guideline, and this has been elucidated in a previous publication from our same research group (Bautista-Orduno K. JCE 2020, doi: 10.1016/j.jclinepi.2020.02.010, PMID: 32145366, https://www.jclinepi.com/article/S0895-4356(19)30836-4/pdf).

Comment #3

The authors encourages panelists on guidelines to fully endorse the GRADE guidance. I do not think this is appropriate in the context of this manuscript.

Response

We agree with the authors perception that the tone of our recommendations is strong. Hence, modifications to lower the tone were made.

---

## [Decision Letter · Decision Letter 1]

26 May 2020

A Deeper Analysis in Thyroid Research: A Meta-Epidemiological Study of the American Thyroid Association Clinical Guidelines

PONE-D-19-35572R1

Dear Dr. Rodriguez-Gutierrez

We are pleased to inform you that your manuscript has been judged scientifically suitable for publication and will be formally accepted for publication once it complies with all outstanding technical requirements.

With kind regards,

Honey V. Reddi

Academic Editor

PLOS ONE

Additional Editor Comments (optional):

We thank the authors for their patience in the time it took for the review of the manuscript. Given the content of the paper, we wanted to ensure it got the most appropriate review.

Kind Regards

Reviewers' comments:

Reviewer's Responses to Questions

**Comments to the Author**

1. If the authors have adequately addressed your comments raised in a previous round of review and you feel that this manuscript is now acceptable for publication, you may indicate that here to bypass the “Comments to the Author” section, enter your conflict of interest statement in the “Confidential to Editor” section, and submit your "Accept" recommendation.

Reviewer #2: All comments have been addressed

2. Is the manuscript technically sound, and do the data support the conclusions?

Reviewer #2: Yes

3. Has the statistical analysis been performed appropriately and rigorously? 

Reviewer #2: Yes

4. Have the authors made all data underlying the findings in their manuscript fully available?

Reviewer #2: Yes

5. Is the manuscript presented in an intelligible fashion and written in standard English?

Reviewer #2: Yes

6. Review Comments to the Author

Reviewer #2: (No Response)

7. PLOS authors have the option to publish the peer review history of their article (what does this mean?). If published, this will include your full peer review and any attached files.

Reviewer #2: No

---

## [Editor Report · Acceptance letter]

29 May 2020

PONE-D-19-35572R1 

A Deeper Analysis in Thyroid Research: A Meta-Epidemiological Study of the American Thyroid Association Clinical Guidelines 

Dear Dr. Rodriguez-Gutierrez:

I am pleased to inform you that your manuscript has been deemed suitable for publication in PLOS ONE. Congratulations! Your manuscript is now with our production department. 

With kind regards,

on behalf of

Dr. Honey V. Reddi 

Academic Editor

PLOS ONE